# Responses of Growth and Root Vitality of *Fokienia hodginsii* Seedling to the Neighbor Competition in Different Heterogeneous Nutrient Environments

Bingjun Li, Mi Deng, Yanmei Pan, Wenchen Chen, Jundong Rong [ID], Tianyou He, Liguang Chen and Yushan Zheng *

College of Forestry, Fujian Agriculture and Forestry University, Fuzhou 350002, China; fafulbj@163.com (B.L.); sweetmi0609@126.com (M.D.); panyanmei@163.com (Y.P.); cerdwin2003@163.com (W.C.); rongjd@126.com (J.R.); hetiantou1885@163.com (T.H.); clguang_cn@163.com (L.C.)
* Correspondence: zys160@163.com

**Abstract:** *Fokienia hodginsii* is characterized by shallow and developed lateral roots with no obvious taproot. It can be used as a pioneer tree species for opening up barren land and as a mixed species for creating plantation forests. During the growth process of *F. hodginsii* seedlings, they are often exposed to heterogeneous nutrient environments and competition for nutrients, water, and space from neighboring plants, which have significant impacts on the growth of *F. hodginsii*. These impacts are most directly reflected in the root vitality of *F. hodginsii*, whose physiological changes embody the plant's adaptability to different heterogeneous nutrient environments and neighbor competition. Herein, high-quality one-year-old *F. hodginsii* seedlings were subjected to three planting patterns to simulate different competition patterns. The three planting patterns were also exposed to three heterogeneous nutrient environments and a homogeneous nutrient environment (control) to determine the differences in the *F. hodginsii* seedling growth and root vitality under different heterogeneous nutrient environments and planting patterns. The *F. hodginsii* seedling height, ground diameter (root neck diameter), and root biomass under the heterospecific neighbor condition were significantly higher than under the single-plant condition. Across heterogeneous nutrient environments, the average seedling height in the N heterogeneous environments was higher than in the P and K heterogeneous environments. The root biomass in the P heterogeneous environment was slightly higher than in the N heterogeneous environment and significantly higher than in the homogeneous and the K heterogeneous environments. Moreover, the average *F. hodginsii* seedling root vitality under the heterospecific neighbor was the highest, and the root vitality in the N heterogeneous environment was the highest under each planting pattern. The *F. hodginsii* root catalase (CAT) and peroxidase (POD) activities under the competition patterns were significantly higher than under the single-plant condition. Additionally, the superoxide dismutase (SOD) activity under single-plant was higher than under the competition patterns. At the same time, the *F. hodginsii* root malondialdehyde (MDA) content was the highest under the conspecific neighbor. Relative to the homogeneous environment, the root CAT, SOD, and POD activities were increased in the N and P heterogeneous environments. The CAT, SOD, and POD activities were highest in the P heterogeneous environment, while the K heterogeneous environment had the highest average MDA content. From the results of principal component analysis, when *F. hodginsii* seedlings were in N and P heterogeneous nutrient environments and heterospecific neighbor, their growth, root biomass accumulation, and root activity indexes reached better levels.

**Keywords:** *Fokienia hodginsii*; heterogeneous nutrient environments; planting patterns; seedling growth; root vitality

## 1. Introduction

Soil is the most important source of plant nutrients. In natural ecosystems, the distribution of soil nutrients is generally heterogeneous due to the significant differences in the texture, moisture content, microbial activity, and decomposition in different forest soils following leaching, fixation, and a series of biochemical processes, which cause significant gradients and patches in the spatial distribution of soil nutrients [1–3]. The spatial heterogeneity of nutrients refers to nutrient changes in the spatial structure of the soil, which are mainly aggregated nutrient environments distributed unevenly or randomly in space [4]. Plants usually adapt to heterogeneous nutrient environments by changing their morphological and physiological plasticity to enhance nutrient absorption from the soil [5–7]. Generally, plants must explore nutrient-rich patches in the soil through continuous root growth to maximize their nutrient acquisition. For this to happen, a series of morphological and physiological root responses guarantee a plant's nutrient acquisition from such patches [8]. As a result, the differences in the physiological plasticity of plant roots often determine the adaptability of plants to different heterogeneous nutrient environments. For example, analysis of the adaptive strategy of *Cunninghamia lanceolata* to heterogeneous nutrient environments revealed that its root proliferated in nutrient-poor patches, and the root P content of *Cunninghamia lanceolata* in nutrient-poor patches was higher than in nutrient-rich patches [9].

Moreover, it has been shown that heterogeneous nutrient environments promoted root development and altered the distribution of nutrient elements in *Pinus massoniana* [10], suggesting that heterogeneous nutrient environments affected root development to varying degrees. In addition, there were significant differences in the tolerance of plant roots to N, P, and K. When the mass concentration of $NH^{4+}$ reached 420 mg/L, which was significantly higher than that of the control group, the growth rate of foxtail stems and the biomass was significantly reduced [11]. Zhao et al. showed that overexpression of the *OsPTR9* gene significantly improved the tolerance of Poplar to low nitrogen stress and increased the mass fraction of the N, P, and K stored in the leaves of Poplar strains [12]. Wang et al.'s study on the molecular mechanism of low-phosphorus tolerance in maize found that *ZmMYB12* was involved in the process of low-phosphorus stress, and it increased the uptake and utilization of phosphorus by plants [13]. It is evident that plant adaptation to different nutrient environments is genetically regulated to show different growth states. In recent years, the responses of plants to the spatial heterogeneity of soil-available resources have gradually become a hotspot in silviculture research. However, existing studies have mostly compared the changes in root morphology and nutrient uptake efficiency [14,15] and rarely compared the differences in the root biomass or activity of plants under different heterogeneous nutrient environments.

Competition is an important phenomenon during plant growth [16–21]. While exploring heterogeneous nutrients, neighboring plant roots inevitably affect plant roots. As a result, the plastic responses of plants to nutrients and neighbor competition play an important role in efficiently utilizing nutrient resources and improving productivity [22]. According to Hodge [23], fast-growing plants take up or deplete most nutrients in the nutrient patches, inhibiting the growth of neighboring plants. However, in silviculture, tree species with high competitiveness can adjust their physiological activity and morphological structure of nutrient organs according to different environments so that their ecological niches are separated, thus better utilizing the nutrient resources in different ecological regions, improving the utilization rate of environmental resources, and realizing the co-existence among species. Generally, plants with higher morphological and physiological plasticity take up nutrients and moisture from the soil faster, promoting their growth and gaining competitive advantages [24]. In forests, most soil nutrients are heterogeneous. The responses of plant roots to neighbor competition significantly vary across different heterogeneous nutrient environments. The competitive relationships between neighboring plants are also more complicated [25,26]. In conclusion, when plants are faced with heterogeneous nutrient environments and competition from neighboring plants, the root

system will change according to different environments and types of competition so that it can better forage for nutrients and water. This trend of changes in the root system and the mechanism of regulation by the two environmental factors in this process have also been a popular research topic in recent years [27].

*Fokienia hodginsii* (Dunn) Henry & Thomas, also known as *Fukien cypress* or *Cupressus hodginsii*, is one of the main mixed species commonly used to create plantations [28]. *F. hodginsii* is moderately photophilous, shade-tolerant when young, and suitable for slightly acidic to acidic yellow and yellow-brown soils. Being resistant to drought and barrenness, it is characterized by shallow and developed lateral roots with no obvious taproot. In China, *F. hodginsii* is mainly distributed in forest lands with an elevation of 350–700 m in southwest, south, and east China [29]. In recent years, studies about *F. hodginsii* have concentrated mainly on cultivation techniques and artificial forest management [30,31]. However, the soil nutrients in the seedling rearing stage are mostly homogeneous, and single-plant is the main planting pattern, which also fails to assess the adaptability and competitiveness of *F. hodginsii* seedlings to nutrient environments in terms of nutrient heterogeneity and inter-species competition. Some previous studies have also demonstrated that different tree species show significant differences in facing heterogeneous nutrient environments and competition from neighboring plants [32–34]; however, there is a lack of research on *F. hodginsii* in this regard. Therefore, in this paper, one-year-old *F. hodginsii* seedlings were selected and assumed to grow in different heterogeneous nutrient environments and encountered different types of neighboring plant competition, whether the growth, root biomass, and activity of *F. hodginsii* seedlings vary, according to the nutrient environment and the planting pattern. Referring to the previous studies, N (nitrogen), P (phosphorus), and K (potassium) heterogeneous nutrient environments were set up and compared with the homogeneous environments. N, P, and K are the most dominant nutrient elements in the soil, and their heterogeneous nutrient environments are most commonly formed in the soil. Neighbor competition is divided into inter-species competition and intra-species competition, using heterospecific neighbor and conspecific neighbor, respectively. Existing research has shown that *Cunninghamia lanceolata* is one of the most suitable mixed species for *F. hodginsii* [35]. Therefore, the inter-species competition is to choose *Cunninghamia lanceolata* as a heterospecific neighbor with *F. hodginsii* for the intra-specific competition, and two *F. hodginsii* plants were planted as a conspecific neighbor, while single-plant were used for comparison. The results of this paper can provide scientific and technical measures and a theoretical basis for the cultivation of *F. hodginsii* seedlings.

## 2. Materials and Methods

### 2.1. Experimental Site

This study was conducted in a greenhouse at the College of Landscape Architecture, Fujian Agriculture and Forestry University (119°13′51.18″ E, 26°05′4.35″ N). The greenhouse was well-ventilated and equipped with a spray cooling system and shading nets. During the experiment, the overall temperature of the greenhouse during the experiment ranged from 14 to 30 °C, with an average temperature of 25 °C, a relative humidity of >78%, a daily sunshine duration of 6:00–18:00, and an average light duration of about 12 h.

### 2.2. Experimental Materials

High-quality one-year-old clonal *F. hodginsii* seedlings cultivated at the Fengtian State-owned Forest Tree Nursery of Anxi County, Quanzhou, Fujian Province, China, were used in this study. To ensure consistency in the initial state of the *F. hodginsii* seedlings during the experiment, 180 *F. hodginsii* seedlings, with an average ground diameter of $2.65 \pm 0.86$ mm and an average seedling height of $21.47 \pm 4.12$ cm, were selected.

A pot experiment was conducted in four nutrient environments: N, P, and K heterogeneous environments and a homogeneous environment. The potting substrate was barren acidic red loam, with a pH of 4.97 and an organic matter content of 5.91 g·kg$^{-1}$, collected from the hill behind Fujian Agriculture and Forestry University. Its total N and P contents

were 0.43 and 0.40 g·kg$^{-1}$, and its hydrolyzable N, available K, and available P contents were 29.08, 238.68, and 6.15 mg·kg$^{-1}$, respectively.

### 2.3. Experimental Design

A $4 \times 3$ two-factor factorial design (consisting of four nutrient environments and three planting patterns) was adopted for this study. The *F. hodginsii* seedlings were planted in polyethylene pots with an upper inner diameter of 22.3 cm, a lower inner diameter of 15.5 cm, and a height of 20 cm. The experiment started in early March 2022. Firstly, the barren, acidic red loam soil was transported from the hill behind Fujian Agriculture and Forestry University to the greenhouse for plowing. The soil was then disinfected with 0.5% potassium permanganate, covered and sealed with a plastic film, and exposed to the sun for one week for air drying before sieving. After sieving, the soil was mixed with perlite in a mass ratio of 3:1 and used as the matching substrate for creating heterogeneous and homogeneous nutrient environments. The upper part of the container was a buffer layer, and the soil was filled to a depth of 16 cm and divided into three parts, namely, the nutrient-rich patch (A side), the nutrient-poor patch (B side), and the middle nursery planting area (This part is the initial growing area of the seedlings, where the soil is potting substrate soil and no fertilizer has been applied). The two nutrient patches had the same volume and were separated from the nursery planting area by a non-woven fabric coated with agar. The agar-coated non-woven material prevents nutrient runoff and loss between the two plots, which affects the test results, and ensures that the root system of *F. hodginsii* seedlings can penetrate smoothly [9] so that it is convenient to observe the growth of the root system in the later stage. Each pot was filled with around 4.5 kg of soil (Figure 1).

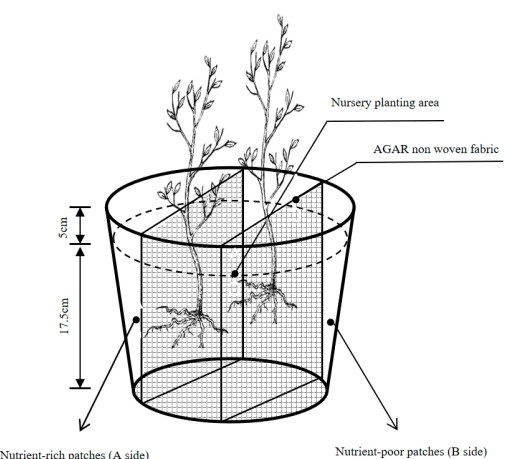 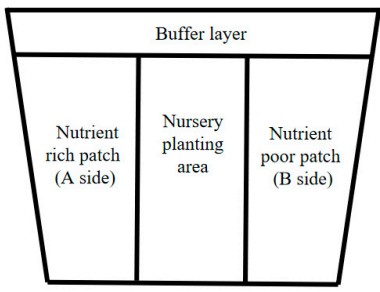

**Figure 1.** Design drawing of pot container.

Nutrient formulations for the heterogeneous/homogeneous environments were based on the cultivation of *F. hodginsii* by reference to previous researchers. Since there are fewer studies on *F. hodginsii*—but only on N heterogeneity—we combined the studies of the nutrient heterogeneity of *Masson pine* and *Cunninghamia lanceolata* [32,34] and the ratios of N, P, and K fertilization. According to the nutrient ratios proposed by Mou [35], the three main nutrient ratios were N:P:K = 2:5:3. It is concluded that the concentration of the N element in the fertilization of *F. hodginsii* in the seedling stage should not be more than 110 mg·kg$^{-1}$, the concentration of the P element should not be more than 300 mg·kg$^{-1}$, and the concentration of the K element should not be more than 190 mg·kg$^{-1}$.

Therefore, when constructing homogeneous nutrient patches, the nutrient patches on both sides of the setup could probably have N, P, and K concentrations of 50, 125, and 75 mg·kg$^{-1}$. An amount of 1 kg of the substrate was mixed with 0.1087 g of urea (N 46%), 1.0081 g of calcium superphosphate (P$_2$O$_5$ 16%), and 0.1433 g of potassium chloride (K$_2$O 60%) on both sides of the container. To construct heterogeneous nutrient environments for different nutrient elements, one side of the container was filled with

enriched soil for the corresponding nutrient element, and twice the fertilizer corresponding to that element was applied so that the corresponding element content was twice as much as that of the homogeneous environments, whereas the depleted environments do not have the corresponding nutrient element applied. For example, in the N heterogeneous nutrient environments, the enriched environments applied twice the amount of urea per kilogram of soil (0.2174 g), whereas for the depleted environments on the other side, urea was not applied to ensure that the total amount of N, P, and K nutrients added to the heterogeneous and homogeneous nutrient environments were the same. The specific fertilization regime is shown in Table 1. In treatments with different heterogeneous nutrient environments, the A side represents the nutrient-rich side of the corresponding nutrient element in the heterogeneous environment, and the B side represents the nutrient-rich side. In treatments with a homogeneous nutrient environment, the contents of each nutrient element on both sides were the same.

**Table 1.** N, P, and K concentrations in heterogeneous and homogeneous nutrient environments.

| Nutrient Environments | Nutrient Patches | | | | | |
|---|---|---|---|---|---|---|
| | A (Nutrient-Rich Patch) | | | B (Nutrient-Poor Patch) | | |
| | N | P | K | N | P | K |
| HET-N | 100 | 125 | 75 | 0 | 125 | 75 |
| HET-P | 50 | 250 | 75 | 50 | 0 | 75 |
| HET-K | 50 | 125 | 150 | 50 | 125 | 0 |
| HOM | 50 | 125 | 75 | 50 | 125 | 75 |

Additionally, three planting patterns were adopted in the heterogeneous and homogeneous nutrient environments, including the single-plant, the *F. hodginsii-F. hodginsii* planting pattern as a conspecific neighbor, and the *F. hodginsii-Cunninghamia lanceolata* planting pattern as a heterospecific neighbor (*Cunninghamia lanceolata,* as the most common and suitable hybrid species for *F. hodginsii*, has been confirmed by most research institutes [36]). One *F. hodginsii* seedling was planted in the middle of each pot for the single-plant condition. For the conspecific neighbor condition, two *F. hodginsii* seedlings were planted at two opposite points on the two sides of the midline, 5 cm away from the midpoint of the midline. For the heterospecific neighbor condition, one *F. hodginsii* seedling and one *Cunninghamia lanceolata* seedling were planted on the two sides of the midpoint of the midline. The *Cunninghamia lanceolata* seedlings were high-quality one-year-old seedlings obtained from the Yangkou State-owned Forest Farm of Fujian Province. The planting depths were about 5 cm. Each treatment consisted of 15 pots, which is 180 pots in total. About 100 mL of purified water was applied daily per pot. Considering the long growth cycle, fertilization was conducted again in September 2022 and February 2023 to maintain the nutrient environments. The nutrient formulae and fertilization rates were the same as those in the first fertilization during planting.

### 2.4. Index Measurement

The seedlings were harvested in April 2023. Ten plants with normal growth were selected from each treatment, and their seedling heights and ground diameters were measured. Additionally, each *F. hodginsii* seedling pot was cut open from the outside with scissors to remove the entire seedling. The soil attached to the roots was rinsed with tap water, followed by secondary cleaning with distilled water. Subsequently, the roots were wiped dry with absorbent paper. Finally, the non-woven fabric was cut open to preserve intact root tissues. To prevent root vigor and antioxidant enzyme activity from being affected, the roots were separated and sealed in self-sealing bags. The bags were then placed in an ice box and transported to the laboratory, where they were immediately processed at low temperatures.

Next, the root tissues of *F. hodginsii* from all the treatments were placed in an ice-filled incubator, and 15 root tips or white roots were sampled from *F. hodginsii* to measure the root vitality and antioxidant enzyme activity. Root vitality was determined using TTC staining [37]. Additionally, the superoxide dismutase (SOD), peroxidase (POD), and catalase (CAT) activities were measured using SOD, POD, and CAT kits (Suzhou COMIN Biotechnology Co., Ltd., Suzhou, China) according to the manufacturer's instructions. The root malondialdehyde (MDA) content was also measured using the thiobarbituric acid (TBA) colorimetric method [38].

Subsequently, the intact roots per treatment were placed in an oven. Firstly, the roots were sterilized by turning the oven temperature to 105 °C for 30 min and then dried at 80 °C to a constant mass to determine the root biomass of *F. hodginsii* under each treatment.

### 2.5. Data Analysis

The SPSS 22.0 software was used for all statistical analyses. All data were analyzed by one-way analysis of variance (ANOVA). Tukey's HSD method was used to determine whether there was a significant difference between the indicators under different treatments ($\alpha$ = 0.05). Additionally, a double-factor variance analysis was conducted to analyze whether the two environmental factors interacted with the root growth, biomass, vitality, and enzyme activity of *F. hodginsii*, thus analyzing the relevance of indicators using the correlation methodology. Finally, principal component analysis (PCA) was employed to identify the dominant factors and comprehensively rank different combinations of environmental factors. The Origin2022 software was adopted for figure preparation.

Principal component analysis (PCA) is a method of multivariate statistical analysis in which multiple variables are linearly transformed to select a smaller number of significant variables. The method mainly includes the following steps: (1) Standardization, where raw data are standardized by subtracting the mean and then dividing by the standard deviation; (2) calculating the covariance matrix by taking the standardized data and finding the covariance matrix between multiple variables; (3) calculating eigenvalues and eigenvectors through eigenvalue decomposition of the covariance matrix; (4) constructing principal components based on eigenvalues and eigenvectors, whereby the number of principal components can be calculated; (5) calculating the composite score for each indicator [39].

## 3. Results

### 3.1. Effects of Planting Patterns on the Growth and Root Biomass of F. hodginsii in Heterogeneous Nutrient Environments

Among the three planting patterns, the *F. hodginsii* seedlings under heterospecific neighbor had the highest seedling height, ground diameter, and root biomass (Figure 2). Specifically, the average seedling height of *F. hodginsii* seedlings in the four nutrient environments reached 27.86 cm under mixed planting, which was 10.7% and 16.7% higher than under conspecific neighbor and single-plant conditions, respectively. Moreover, the differences between the homogeneous and the N heterogeneous environments were significant ($p < 0.05$). The average ground diameter of *F. hodginsii* seedlings under heterospecific neighbor was 3.51 mm, which was 8.6% and 11.5% higher than conspecific neighbor and single-plant, respectively. Additionally, the average ground diameter in the homogeneous and K heterogeneous environments under heterospecific neighbor was significantly higher than under single-plant ($p < 0.05$). The average root biomass of *F. hodginsii* seedlings under heterospecific neighbor reached 3.61 g and was also significantly higher than under the other two planting patterns ($p < 0.05$). Specifically, it was 17.6% higher than under conspecific neighbor and 31.4% higher than under single-plant.

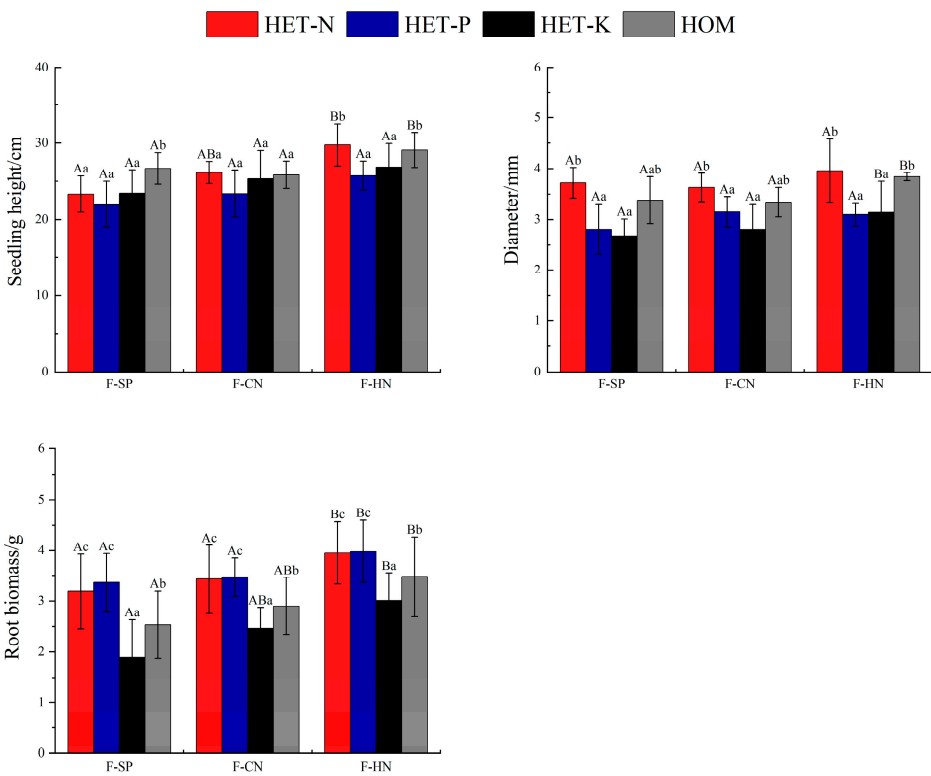

**Figure 2.** Differences in growth and root biomass of *F. hodginsii* seedlings under different nutrient environments and planting patterns. Note: F-SP: single-plant; F-CN: *F. hodginsii-F. hodginsii* conspecific neighbor; F-HN: *F. hodginsii-Cunninghamia lanceolata* heterospecific neighbor. Error bars indicate standard errors. The different capital letters indicate significant differences between different planting patterns in the same nutrient environments; the different lowercase letters indicate significant differences between different nutrient environments under the same planting pattern ($p < 0.05$).

At the same time, the average seedling height of *F. hodginsii* in the homogeneous nutrient environment reached 27.20 cm and was higher than in the three planting patterns in any of the heterogeneous nutrient environments. In addition, the average seedling height in the N heterogeneous environments was 11.2% higher than in the P heterogeneous environments and 4.7% higher than in the K heterogeneous environments. Notably, under heterospecific neighbor, the average seedling heights in the homogeneous and N heterogeneous environments were significantly higher than in the P and K heterogeneous environments ($p < 0.05$).

The average ground diameter of *F. hodginsii* under different planting patterns reached 3.77 mm in the N heterogeneous environments, which was higher but not significantly different than under the homogeneous nutrient environment ($p > 0.05$). Notably, it was 24.8% higher than in the P heterogeneous environments and 31.1% higher than in the K heterogeneous environments, with significant differences under the different planting patterns ($p < 0.05$). The average root biomass of *F. hodginsii* in the P heterogeneous environments reached 3.60 g, slightly higher than in the N heterogeneous environments but significantly higher than in the homogeneous environments (21.3%) and the K heterogeneous environments (46.9%).

According to the two-way ANOVA of the effects of planting patterns and nutrient heterogeneity on the growth and root biomass of *F. hodginsii* seedlings (Table 2), planting patterns and nutrient heterogeneity exerted highly significant effects on the *F. hodginsii* seedling height ($p < 0.01$) but did not have any significant interactive effect. In addition, the planting patterns did not significantly influence the ground diameter of *F. hodginsii* ($p > 0.05$), unlike the nutrient heterogeneity, which exerted a significant effect ($p < 0.05$). Notably, the planting patterns and nutrient heterogeneity both exerted significant effects

on the *F. hodginsii* root biomass, with nutrient heterogeneity having a highly significant effect. Moreover, planting patterns and nutrient heterogeneity had a significant interactive effect on the root biomass.

**Table 2.** Effects of planting patterns and nutrient heterogeneity on growth and root biomass of *F. hodginsii* seedlings.

| | *p* Value | | |
| --- | --- | --- | --- |
| | **Planting Pattern** | **Nutrient Heterogeneous Patches** | **Effects between Planting Patterns and Nutrient Heterogeneous Patches** |
| Seeding height | 0.006 ** | 0.004 ** | 0.058 ns |
| Diameter | 0.063 ns | 0.042 * | 0.279 ns |
| Root biomass | 0.013 * | 0.008 ** | 0.012 * |
| Degrees of freedom | 2 | 3 | 6 |

*, $p < 0.05$; **, $p < 0.01$; ns, $p > 0.05$.

### 3.2. Effects of Planting Patterns on the F. hodginsii Root Vitality in Heterogeneous Nutrient Environments

The average *F. hodginsii* root vitality reached 88.38 $\mu gTTF \cdot g^{-1} \cdot h^{-1}$ under heterospecific neighbor, which was significantly higher than under single-plant (64.6%) and conspecific neighbor (36.7%) (Figure 3). The average root vitality under heterospecific neighbor was also significantly higher than in single-plant and conspecific neighbor in each nutrient environment ($p < 0.05$). Notably, the root vitality in the N environments under heterospecific neighbor peaked compared to all treatments. The mean value of root vigor under the N heterogeneous nutrient environments in each planting pattern amounted to 105.33 $\mu gTTF \cdot g^{-1} \cdot h^{-1}$ and was significantly higher than in the P heterogeneous environments (48.1%), K heterogeneous environments (135.9%), and the homogeneous environments (93.1%). Clearly, the root vitality was improved in the N and P heterogeneous environments relative to the homogeneous environment, while the K heterogeneous environment inhibited root vitality to some extent.

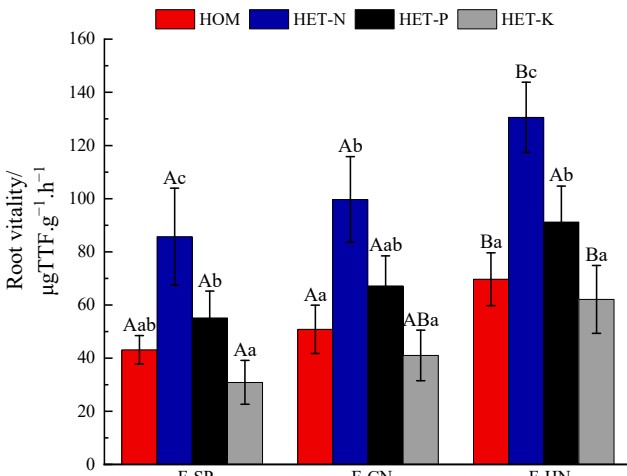

**Figure 3.** Differences in the root vitality of *F. hodginsii* seedlings under different treatments. Note: F-SP: single-plant; F-CN: *F. hodginsii-F. hodginsii* conspecific neighbor; F-HN: *F. hodginsii-Cunninghamia lanceolata* heterospecific neighbor. Error bars indicate standard errors. The different capital letters indicate significant differences between different planting patterns in the same nutrient environments; the different lowercase letters indicate significant differences between different nutrient environments under the same planting pattern ($p < 0.05$).

Moreover, the planting patterns and nutrient heterogeneity exerted significant effects on the root vitality of *F. hodginsii* ($p < 0.05$), with the planting patterns exerting a highly significant effect (Table 3). Additionally, the two had a significant interactive effect on root vitality.

**Table 3.** Effects of planting patterns and nutrient heterogeneity on root vitality of *F. hodginsii* seedlings.

| | *p* Value | | |
| --- | --- | --- | --- |
| | **Planting Pattern** | **Nutrient Heterogeneous Patches** | **Effects between Planting Patterns and Nutrient Heterogeneous Patches** |
| Root vitality | 0.006 ** | 0.026 * | 0.041 * |
| Degrees of freedom | 2 | 3 | 6 |

*, $p < 0.05$; **, $p < 0.01$.

*3.3. Effects of Planting Patterns on the Root Enzyme Activity of F. hodginsii in Heterogeneous Nutrient Environments*

As can be seen from Figure 4, the average root CAT activity in *F. hodginsii* was the highest under heterospecific neighbor, reaching as high as 64.29 nmol·min$^{-1}$·g$^{-1}$, which was significantly (41.7%) higher than under single-plant ($p < 0.05$). At the same time, the average root CAT activity under conspecific neighbor was 57.38 nmol·min$^{-1}$·g$^{-1}$, 26.5% higher than under single-plant. The root POD activity of *F. hodginsii* presented a trend similar to that of the root CAT activity under each planting pattern. Specifically, the average root POD activity was the highest under heterospecific neighbor, reaching as high as 1815.88 U·g$^{-1}$, 70.7% and 35.3% higher than under the single-plant and conspecific neighbor conditions, respectively. Notably, there were significant differences in the POD and CAT activities among the three planting patterns in the N and P nutrient environments.

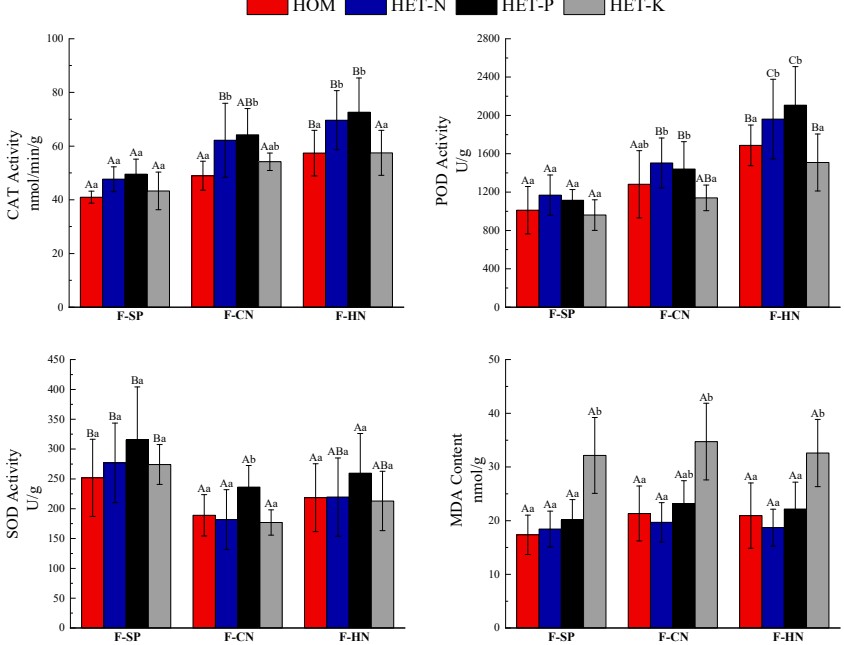

**Figure 4.** Differences in enzyme activity in *F. hodginsii* seedlings under different treatments. Note: F-SP: single-plant; F-CN: *F. hodginsii-F. hodginsii* conspecific neighbor; F-HN: *F. hodginsii-Cunninghamia lanceolata* heterospecific neighbor. Error bars indicate standard errors. The different capital letters indicate significant differences between different planting patterns in the same nutrient environments; the different lowercase letters indicate significant differences between different nutrient environments under the same planting pattern ($p < 0.05$).

In contrast, the average *F. hodginsii* root SOD activity was the highest under single-plant, reaching as high as 279.51 $U \cdot g^{-1}$, 22.8% and 42.6% higher than under heterospecific neighbor and conspecific neighbor, respectively. Moreover, the root SOD activity under single-plant was significantly higher than under the heterospecific neighbor and conspecific neighbor conditions in the four nutrient environments ($p < 0.05$). On the other hand, the root MDA content was the highest under conspecific neighbor, which was 4.8% higher than under heterospecific neighbor, and single-plant had the lowest root MDA content (22.02 $nmol \cdot g^{-1}$). However, the differences in the MDA content among the three planting patterns were not significantly different ($p > 0.05$), implying that neighbor competition increased the root MDA content of *F. hodginsii* to some extent, though not significantly.

Analysis of the effects of nutrient environments on the root antioxidant enzyme activity and MDA content in *F. hodginsii* roots revealed that the CAT activity was the highest in the P nutrient environments across all three planting patterns, with an average of 62.13 $nmol \cdot min^{-1} \cdot g^{-1}$, which was significantly (26.5%) higher than in the homogeneous environment ($p < 0.05$). The CAT activity was second highest in the N nutrient environments. Additionally, the CAT activity in the K nutrient environments was slightly higher than in the homogeneous environment, but the difference was non-significant ($p > 0.05$).

The POD activity changes were similar to the CAT activity in each nutrient environment. Specifically, the average POD activity was the highest in the P environments, reaching as high as 1553.57 $U \cdot g^{-1}$, 17.1% higher than in the homogeneous environments. In addition, the average POD activity in the N environments was slightly lower than in the P environments, but the difference was insignificant. Similarly, the average POD activity in the K environments was 10.2% lower than in the homogeneous environments. These results revealed that the N and P heterogeneous environments enhanced the root CAT and POD activities of *F. hodginsii* relative to the homogeneous environment. On the other hand, the differences between the K heterogeneous and homogeneous environments were insignificant. The nutrient heterogeneous environments with the highest SOD activity were the P environments, with an average of 270.37 $U \cdot g^{-1}$, 23.1% higher than the homogeneous environments. Moreover, the SOD activity in the N and K environments was slightly higher than in the homogeneous environments, though the differences were insignificant ($p > 0.05$). Moreover, the nutrient heterogeneous environments with the lowest MDA content were the N environments, with an average of 18.94 $nmol \cdot g^{-1}$. However, there were no significant differences in the MDA content between the N and P environments or with the homogeneous environments ($p > 0.05$). On the other hand, the MDA content in the K environments (33.16 $nmol \cdot g^{-1}$ on average) was significantly higher than in any other environments, implying that the K heterogeneous environment significantly increased the *F. hodginsii* root MDA content.

Analysis of the root antioxidant enzyme activity and MDA content in *F. hodginsii* seedlings revealed that the planting patterns and nutrient heterogeneity both exerted highly significant effects on the root CAT and SOD activities ($p < 0.01$), with a significant interactive effect on the activity of the two enzymes ($p < 0.05$) (Table 4). Similarly, the planting patterns and nutrient heterogeneity also exerted significant effects on the *F. hodginsii* root POD activity, with the nutrient heterogeneity exerting a highly significant effect. However, their interactive effect on the root POD activity was insignificant. The effect of planting patterns on the MDA content was also insignificant, while that of nutrient heterogeneity was significant. Their interactive effect on the MDA content was also insignificant.

**Table 4.** Effects of planting patterns and nutrient heterogeneity on enzyme activity in *F. hodginsii* seedlings.

| | *p* Value | | | *F* Value | | |
|---|---|---|---|---|---|---|
| | Planting Pattern (a) | Nutrient Heterogeneous Patches (b) | Effects between a and b | Planting Pattern (a) | Nutrient Heterogeneous Patches (b) | Effects between a and b |
| CAT | 0 ** | 0 ** | 0.045 * | 19.568 | 21.869 | 6.002 |
| POD | 0.015 * | 0 ** | 0.609 ns | 6.117 | 16.323 | 0.751 |
| SOD | 0.001 ** | 0 ** | 0.033 * | 12.795 | 22.004 | 3.327 |
| MDA | 0.853 ** | 0.026 * | 1.375 ns | 0.428 | 3.056 | 0.225 |
| Degrees of freedom | 3 | 2 | 6 | 3 | 2 | 6 |

*, $p < 0.05$; **, $p < 0.01$; ns, $p > 0.05$.

### 3.4. Correlation between F. hodginsii Growth and Root Indexes

The *F. hodginsii* growth and root development were affected by multiple factors and changes in environmental factors, which caused the differences in its growth, root biomass, root vitality, and root enzyme activity. There were also interactions between various indexes. For example, the seedling height was highly significantly positively correlated with the ground diameter. The seedling height and ground diameter were significantly positively correlated with the root biomass (Table 5). The root biomass was also significantly positively correlated with root vitality. Notably, root vitality had a significant positive correlation with the activity of each of the three antioxidant enzymes, with its correlations with POD and SOD activities reaching highly significant levels.

**Table 5.** Correlations between *F. hodginsii* seedling growth and root indexes.

| Parameter | Seedling Height | Diameter | Root Biomass | Root Vitality | CAT | POD | SOD | MDA |
|---|---|---|---|---|---|---|---|---|
| Seedling height | 1 | | | | | | | |
| Diameter | 0.681 ** | 1 | | | | | | |
| Root biomass | 0.249 * | 0.254 * | 1 | | | | | |
| Root vitality | 0.017 | −0.035 | 0.258 * | 1 | | | | |
| CAT | 0.092 | 0.043 | 0.175 | 0.253 * | 1 | | | |
| POD | 0.203 | 0.208 | 0.141 | 0.461 ** | 0.262 * | 1 | | |
| SOD | 0.156 | 0.112 | 0.175 | 0.449 ** | 0.490 ** | 0.348 ** | 1 | |
| MDA | 0.147 | −0.087 | −0.227 * | −0.288 ** | −0.186 | −0.200 | −0.102 | 1 |

*, $p < 0.05$; **, $p < 0.01$.

On the contrary, root vitality had a highly significant negative correlation with the MDA content. Moreover, the CAT activity had a significant positive correlation with POD activity. Additionally, the CAT and POD activities were highly significantly positively correlated with SOD activity. Notably, the activity of each of the three antioxidant enzymes was negatively correlated with the MDA content, but the correlation was insignificant.

### 3.5. Comprehensive Evaluation of the F. hodginsii Growth and Root Indexes

PCA revealed that the cumulative variance contribution rate of the first three principal components reached 91.433%, implying that they could explain all the data variabilities (Table 6). Therefore, the first three principal components were adopted as the comprehensive evaluation indexes. Notably, the absolute values of the coefficients of root biomass, root vitality, POD activity, and ground diameter in the first principal component were large, suggesting that they were the most important physiological indexes influencing the growth and root development of *F. hodginsii* seedlings in different heterogeneous nutrient environments and under different planting patterns.

**Table 6.** PCA based on the growth and root indexes of *F. hodginsii* seedlings.

| Parameter | Principal Component | | |
|---|---|---|---|
| | 1 | 2 | 3 |
| Root biomass | 0.941 | −0.175 | −0.201 |
| Root vitality | 0.879 | −0.133 | 0.294 |
| POD | 0.848 | 0.234 | 0.42 |
| Diameter | 0.847 | −0.300 | −0.365 |
| Seedling height | 0.760 | 0.312 | −0.400 |
| CAT | 0.724 | 0.356 | 0.568 |
| MDA | −0.532 | 0.793 | 0.074 |
| SOD | −0.378 | −0.698 | 0.486 |
| Eigenvalue | 4.623 | 1.532 | 1.159 |
| Contribution rate/% | 57.79 | 19.151 | 14.492 |
| Cumulative contribution rate/% | 57.79 | 76.941 | 91.433 |

Based on the eigenvalues and eigenvectors of the three principal components, one can derive three principal component expressions, which are:

$$Y_1 = 0.438X_1 + 0.409X_2 + 0.394X_3 + 0.394X_4 + 0.353X_5 + 0.337X_6 - 0.247X_7 - 0.176X_8$$

$$Y_2 = -0.141X_1 - 0.107X_2 + 0.189X_3 - 0.242X_4 + 0.252X_5 + 0.288X_6 + 0.641X_7 - 0.564X_8$$

$$Y_3 = -0.187X_1 + 0.273X_2 + 0.390X_3 - 0.339X_4 - 0.371X_5 + 0.527X_6 + 0.069X_7 + 0.451X_8$$

Afterward, the standardized raw data were substituted into the principal component expression to calculate the composite principal component score for each treatment. Comprehensive scores, based on the eigenvalues in the PCA, were highest in HET-N × F-HN, HOM × F-HN, and HET-P × F-HN (Table 7). Notably, the growth and root indexes of the *F. hodginsii* seedlings in the N heterogeneous environment were superior to those in the homogeneous and P and K heterogeneous environments. Additionally, the K heterogeneous environment had poor effects on the growth and root development of *F. hodginsii* seedlings. Overall, the heterospecific neighbor pattern was more conducive to the growth and root development of *F. hodginsii* seedlings than the conspecific neighbor and single-plant patterns.

**Table 7.** A comprehensive evaluation of the treatment effects on growth and root indexes of *F. hodginsii*.

| Treatments (Planting Pattern × Heterogeneous Nutrient Environments) | Comprehensive Score | Comprehensive Rank |
|---|---|---|
| HOM × F-SP | −0.45 | 9 |
| HOM × F-CN | −0.07 | 6 |
| HOM × F-HN | 1.39 | 2 |
| HET-N × F-SP | −0.01 | 5 |
| HET-N × F-CN | 0.87 | 4 |
| HET-N × F-HN | 2.36 | 1 |
| HET-P × F-SP | −1.3 | 10 |
| HET-P × F-CN | −0.21 | 8 |
| HET-P × F-HN | 1.15 | 3 |
| HET-K × F-SP | −2.24 | 12 |
| HET-K × F-CN | −1.32 | 11 |
| HET-K × F-HN | −0.17 | 7 |



## 4. Discussion

The uneven spatial distribution of forest soil nutrients makes the nutrient environment highly heterogeneous, and there are often large differences in the growth processes of different trees in different heterogeneous nutrient environments. The trees also need to face spatial and nutrient competition from neighboring species, which causes plant growth and root development in the different heterogeneous nutrient environments and neighboring competition with significant differences and changes in trends. As a result, forest tree growth and root development significantly vary across the different heterogeneous nutrient environments and neighbor competition, leading to different change trends. For example, the difference in root vitality leads to differences in the competitiveness and productivity of forest trees [40,41]. In this study, the seedling height, ground diameter, and root biomass of *F. hodginsii* seedlings under the heterospecific neighbor condition were significantly higher than those under the single-plant condition, suggesting that the *F. hodginsii-Cunninghamia lanceolata* heterospecific neighbor pattern improved the growth and root biomass accumulation of *F. hodginsii*.

Similarly, Yao [34] also revealed that the growth and root biomass accumulation of *Schima superba* under the heterospecific neighbor condition was significantly higher than under single-plant and conspecific neighbor conditions, regardless of homogeneous or heterogeneous soil nutrient environments. Neighbor competition is either inter-specific or intra-specific. Herein, the seedling height, ground diameter, and root biomass of *F. hodginsii* under heterospecific neighbor were higher than under conspecific neighbor, probably because the neighbor competition induced changes in the plant behavior, including root growth and biomass accumulation [42]. Generally, the interactions between plant roots vary, depending on whether inter-specific or intra-specific competition is involved [43,44] due to the self/non-self-recognition of roots. Precisely, when plant roots encounter non-self-roots, they produce more roots than when encountering self-roots. Similarly, when a plant competes with plants of the same species for limited space and resources, it grows less root biomass than when competing with plants of different species, mainly due to a mutual inhibitory effect. This is known as the "kin recognition effect", in which root secretions are used as signaling substances to distinguish the genetic identity of competitors from neighboring plants and to respond differently [45,46]. According to Hamilton's kin selection theory, plant root systems may tend to reduce foraging behavior when they encounter genetically similar or equivalent neighboring plants [47]. Therefore, *F. hodginsii* under heterospecific neighbor will obtain higher growth efficiency and biomass accumulation due to the expansion of the root system and higher foraging behavior, resulting in higher seedling height, diameter, and root biomass than conspecific neighbor and single-plant patterns.

In this study, the seedling height and ground diameter of *F. hodginsii* were higher in the N heterogeneous environment and the homogeneous environment. However, the root biomass was the highest in the P heterogeneous nutrient environments. Additionally, the root biomass of *F. hodginsii* was significantly increased in the P and N heterogeneous environments relative to the homogeneous and K heterogeneous environments. These results were similar to the findings of Ma [32] and Drew [48]. This may be because of poor P mobility in the soil. Roots need to proliferate and elongate deeper into the soil to acquire sufficient P. Herein, the N heterogeneous environment induced the proliferation of *F. hodginsii* fibrous and lateral roots, enhancing their uptake of soil nutrients, which improved their ability to utilize environment nutrients. However, since N easily diffuses in the soil more than P, the plant roots have limited proliferation when acquiring N from the environments than when acquiring P. This explains why the *F. hodginsii* root biomass in the P environments was higher than in the N environments. Moreover, the *F. hodginsii* root biomass in the K environments was lower than in other environments, similar to the findings of Jin [49] and Guo [50]. This is probably because the high K ion content in the plant during the seedling stage inhibits the uptake of other nutrient elements, resulting in delayed growth and inhibited root biomass accumulation [51].

In addition, planting patterns and nutrient heterogeneity had a significant interactive effect on the *F. hodginsii* root biomass but not on seedling height and ground diameter, implying that the two environmental factors had a superimposed effect only on the *F. hodginsii* root biomass accumulation. This may be related to root extension foraging, the allocation and regulation of assimilation products, and other timely adjustment strategies of neighboring plants with competitive relationships in heterogeneous environments [45,46].

Root vitality is a major index of plant growth [52], which reflects the root metabolism intensity. Generally, the higher the root vitality, the stronger the root nutrient uptake capacity. In this study, the *F. hodginsii* root vitality was significantly higher under neighbor competition than under single-plant. Similarly, Hodge [23] revealed that when *Lolium perenne* was cultivated under heterospecific neighbor conditions with *Poa pratensis*, the root growth rate and root vitality of *Lolium perenne* in nutrient environments were significantly higher than under single-plant, which is consistent with the results in this study. This suggests that plants in a competitive environment gain a competitive advantage over neighboring plants by increasing their root metabolism intensity through enhanced root vitality, which enhances their competitiveness for soil resources, improving their nutrient uptake capacity. Herein, the root vitality was enhanced in the N and P environments relative to the homogeneous environments. However, the root vitality in the K environments was lower than in the homogeneous environments, implying that the K heterogeneous environment caused an imbalance in the K ion concentration in *F. hodginsii* during the seeding stage, thereby affecting its metabolism and, to some extent, inhibiting its root vitality. In addition, the planting patterns and nutrient heterogeneity had a significant interactive effect on the *F. hodginsii's* root vitality, where the root vitality was co-regulated by planting patterns and nutrient heterogeneity, and the root vitality magnitude rested on their combined action.

SOD, POD, and CAT play important roles in plants, mainly by protecting the internal environment mechanism of plants against external stress and plant cells against damage from reactive oxygen species. They also effectively improve the plant's adaptability and resistance to environmental stress; thus, their activity often serves as a physiological and biochemical index of plant aging [53,54]. In this study, the *F. hodginsii* root CAT and POD activities were significantly higher under neighbor competition than in single-plant conditions. They were also higher under the heterospecific neighbor than under conspecific neighbor. Moreover, the effect of the planting patterns on CAT and POD activities reached a highly significant and significant level, respectively. This suggests that CAT and POD activities depend on the intensity and form of neighbor competition to some extent, similar to the changes in root biomass and root vitality. When the *F. hodginsii* roots encounter neighbor competition during foraging, they elevate their metabolism to enhance their competitiveness. However, as the metabolism level rises, excess hydrogen peroxide is generated, causing damage to the plants. Hydrogen peroxide is rapidly converted into other harmless or less toxic substances to avoid such damage. CAT is a tool often used by cells to catalyze the decomposition of hydrogen peroxide, while POD has a dual effect of eliminating the toxicity of hydrogen peroxide, phenols, and amines. Therefore, the root CAT and POD activities are significantly enhanced under competition patterns to eliminate the harmful substances generated by elevated metabolism levels in plants [55].

Moreover, the SOD activity under the single-plant condition was higher than under neighbor competition patterns, possibly due to the lower root CAT and POD activities and less $H_2O_2$ generated under single-plant. To maintain a low level of free radicals in cells and prevent the cell membrane damage caused by free radicals, plants increase the SOD activity to catalyze the dismutation of superoxide anions, thereby generating $H_2O_2$ and $O_2$ [56]. In addition, the *F. hodginsii* root MDA content was the highest under the conspecific neighbor. The MDA content under the heterospecific neighbor was also slightly higher than in the homogeneous batch, though the difference was insignificant. MDA is a plant-membrane lipid peroxidation product, and its content reflects the damage to the cell membrane system [57]. Apparently, when plants of the same species are under neighbor

competition, the same species of plants, due to the similarity of demand for nutrients, space, and ecological niche, will more likely lead to the growth of the plant-encountered nutrient and space resources shortage of adversity; therefore the MDA content in the root system increases. When neighboring plants compete with heterospecific plants and are not dominant species relative to the target plant, it can effectively increase the plant's utilization efficiency of nutrients and space resources in the soil, reduce the generation of adversity, and lower the MDA content in the root system.

The *F. hodginsii* root's CAT, SOD, and POD activities were increased in the N and P heterogeneous environments relative to the homogeneous environment. The P heterogeneous environments had the highest CAT, SOD, and POD activities, though the differences between the P and the N heterogeneous environments were non-significant, similar to the findings by Chen [27]. N and P are essential nutrient elements for plants. In the N and P heterologous environments, plants have some roots in the nutrient-poor environments, where they cannot easily acquire sufficient nutrients. This stimulates plants to transfer inducing signals, increasing plant roots in abundance and enhancing antioxidant enzyme activity to support a series of metabolic activities. By contrast, the difference in the *F. hodginsii* root antioxidant enzyme activities between the K heterogeneous and homogeneous environments was insignificant. Possibly, *F. hodginsii* has a low tolerance of the K heterogeneous environment, which has a certain inhibitory effect on its antioxidant enzyme activity, which limits its antioxidant enzyme activity in the homogeneous environment than in the N and P heterologous environments. The N heterogeneous environments had the lowest MDA content, though it is insignificantly different from the P and homogeneous environments. On the other hand, the MDA content in the K heterogeneous environments was significantly higher than the other nutrient environments, implying that *F. hodginsii* had a low tolerance to the K heterogeneous environment, which intensified the cell damage, making it impossible to adapt well to the environment.

The correlation analysis also revealed that *F. hodginsii's* root vitality had a significant positive correlation with the root biomass, seedling height, ground diameter, and the activity of the three antioxidant enzymes, suggesting that the level of root vitality reflects the root biomass accumulation, growth status, and the activity of antioxidant enzymes in plant roots. SOD, POD, and CAT activities were significantly positively correlated, yet negatively correlated with the MDA content, though the correlation was insignificant. Moreover, the absolute values of the coefficients of root biomass, root vitality, POD activity, and ground diameter were significantly higher than those of other indexes, suggesting that these physiological indexes are relatively sensitive in a growth environment with different planting patterns and heterogeneous nutrient environments, and can satisfactorily reflect the growth and root vitality of *F. hodginsii* seedlings. Therefore, based on the calculated comprehensive scores, according to the eigenvalues of the PCA patterns with high comprehensive scores, including HET-N × F-HN, HOM × F-HN, and HET-P × F-HN, these could support enhanced *F. hodginsii* growth and root vitality.

## 5. Conclusions

In this study, the growth, root biomass, root vigor, and enzyme activities of *F. hodginsii* seedlings under different heterogeneous nutrients and planting patterns were analyzed and evaluated. *F. hodginsii's* seedling height and ground diameter are higher in the N heterogeneous environments than in other heterogeneous nutrient environments and homogeneous environments. Moreover, the N and P nutrient environments significantly support increased root biomass and root activity of *F. hodginsii* seedlings and the activity of the three antioxidant enzymes. The K heterogeneous environment adversely influences *F. hodginsii's* seedling growth and root development. Notably, the heterospecific neighbor is more conducive for the *F. hodginsii* seedling growth and root development than the conspecific neighbor and single-plant patterns. *F. hodginsii* is a common barren-tolerant tree species with high adaptability, which is the first choice for afforestation and companion species. The activity and growth differences of root systems in different heterogeneous

nutrient environments have not been reported yet, whereas *Cunninghamia lanceolata* is a common hybrid and symbiotic species in the *F. hodginsii* branches. There is a gap in the cultivation of *F. hodginsii* species regarding whether the heterospecific neighbors of *F. hodginsii* and cedar can promote root growth and activity in the seedling stage. Therefore, this study is based on these two aspects of research as the background and foundation of the experiment, which can provide a valuable theoretical basis for the subsequent cultivation of *F. hodginsii* seedlings.

Moreover, the molecular mechanisms underlying the foraging ability of *F. hodginsii* and its competitiveness in neighbor competition are yet to be clarified. Thus, further efforts are still needed to complete research in this direction. Future studies should explore the responses of the morphological or anatomical root structures to nutrient environments and planting patterns and reveal the mechanisms governing the responses of *F. hodginsii* seedlings to heterogeneous nutrient environments and competition patterns at the molecular level.

**Author Contributions:** Software, Y.P.; Validation, W.C.; Investigation, M.D.; Resources, J.R.; Writing—original draft, B.L.; Supervision, L.C.; Project administration, Y.Z.; Funding acquisition, T.H. All authors have read and agreed to the published version of the manuscript.

**Funding:** This research was funded by Fujian Seedling Science and Technology Research Project (LZKG-202207) and Forestry Peak Discipline Construction Project from Fujian Agriculture and Forestry University (72202200205).

**Data Availability Statement:** Data are contained within the article.

**Conflicts of Interest:** The authors declare no conflict of interest.

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
