# Peer review of "Responses of Growth and Root Vitality of Fokienia hodginsii Seedling to the Neighbor Competition in Different Heterogeneous Nutrient Environments"

_forests, doi:10.3390/f14122278_

Round 1

Reviewer 1 Report

Comments and Suggestions for Authors

The MS entitled "Responses of growth and root vitality of Fokienia hodginsii seedling to the neighbor competition in different heterogeneous nutrient environments " is a nice piece of work done by the authors.
The application of nitrogen for agricultural purposes is one of the significant activities responsible for the disruption of natural ecosystems and environmental problems globally.

This paper focuses on the examination of the effect of N, P, and K applications on different growth characteristics and enzymatic activity in tree seedlings.

The science structure and overall theme of the manuscript are sound and acceptable.  In my opinion, the overall concept is interesting and important.

The results suggest some differences in nutrient incorporation and dependent enzymic activity. However, a large variability is evident (see ANOVA in figures). PCA analysis is reported in the Methods (poorly explained) and moreover is not presented anywhere in the results thereafter. The conclusions do not represent causality, but a further advance in theoretical research for practical cultivation.

The paper requires only Minor revisions to be acceptable for publication, and I have a few minor suggestions which I believe will improve the manuscript.
The abstract can serve as a stand-alone document that succinctly describes both procedures and conclusions. In general, the paper is clear and well-written.
My suggestion regarding this MS is to accept after revising with the following points:

1. The authors presented a very good introduction, but they could explain more about the molecular mechanisms of plant N, P, K tolerance and potential mitigation from N, P, K toxicity.

2. Introduction and discussion sections authors could also involve new aspects - hormonal regulation of the N, P, K tolerance (interaction with SA, CYT etc), including new references.

3. Add more view on photosynthetic parameters/discuss which kind of parameters are more sensitive and why.

4. Add some recent references in the MS. It is important to discuss the plant regulation mechanisms.

5. The manuscript also requires additional editing to for grammar and proper word use for the English language. It is well written but does require some additional work prior to final publication.

The paper brings many new aspects, and the novelty of the paper is OK, but I would like to invite authors to discuss also more eco-physiological aspects using new references: DOI: 10.17957/IJAB/15.1578; 10.1111/jvs.13052; 10.1016/j.sjbs.2017.11.049; 10.3390/agronomy11030448; 10.1016/j.scienta.2016.07.032

Final COMMENTS
- The manuscript is useful and innovative; it contains original data.
- This study presents the relevant matter in more depth than some of the other related publications; therefore, I recommend its publication after MODERATE  changes.

Author Response

Thank you very much for your valuable comments, in response to your comments, I have made the following changes to the MS: 1, in the research methodology and analysis of the results added the interpretation of the principal component analysis and the calculation process. 2, in the introduction part of the MS added the current progress of the research on the tolerance of plants to nutrient elements and some of the molecular mechanisms. 3, improved the overall structure of the MS and the language of professionalism and completeness

Reviewer 2 Report

Comments and Suggestions for Authors

Overall comments

The manuscript provides an interesting set of results from a well-designed study to determine the morphological and physiological responses to neighbor competition and soil environment. The authors did a fantastic job at explaining the theoretical backgrounds for the study and discussing the observed results. Some of the passages are a bit long and can be a bit more concise. I have provided some comments that should be useful for improving your manuscript.

Specific comments

Abstract
  • The abstract is a bit long. Consider shorten it.
  • Would be nice to briefly mention what this plant species is, and why the reader should be interested in their growth and root vitality.
  • L.15-16: The treatments can be a bit confusing. Probably use the following terms to be consistent with other studies in this field: "single-plant", "conspecific neighbor", "heterospecific neighbor". Or later in your manuscript, you use "inter-species" and "intra-species." Should be consistent.
  • L.40-43: Not clear this means. PCA is not exactly set up to test the correlations as described here
  • The end of the introduction should summarize the effects of neighbor competition and nutrient environment.

Introduction

  • L.92: The genus name should be Cupressus. Also the synonym of Fokienia hodginsii is actually Cupressus hodginsiiCupressus duclouxiana is a different species, accoring to the Plant of the World Database 
  • L.110: Citations to the previous studies should be provided here.
  • L.115: Citations to the existing research should be provided here. Also what does it mean for C. lanceolata to "one of the most sutiable mixed species for F. hodginsii?

Materials and Methods

  • Please make sure that 
    • scientific names are italicized throughout. 
    • double periods (..) at the end of some sentences are removed.
    • a space exists after the peroid of each sentence.
  • L.154: what does it mean by "4cmT"?
  • L.165: citation for the previous reference here.
  • L.166: In-text citation should be numbered instead of parenthetical.
  • L.171: Provide the citation for Mou.
  • Table 1 is a bit confusing. Homogenous should be listed as the fourth row in the table, so the readers understand immediately that there were four treatments for soil envioronment.
  • L.242: Not sure if this is the best use of PCA, because PCA moslty attempts to find the best linear combinations for the existing multivariate data. Multiple regressions might be more appropriate for finding dominant factors?

Results

  • Figure 2 is a bit hard to me personally. I understand that the authors want to convey as much information as possible in one set of graphs. My preference is to see each graph (or set of graph) for each measurement separately (i.e. one y-axis for each graph only). Similar format to Figures 3 and 4.
  • L.283: Not clear what it means by "the same below."
  • Table 2/3/4: F-values/Degree of freedom might not be necessary to report in the main text. Also "Effect between a and b" is probably better phrased as "interaction." Consider not using "a" and "b" so that it doesn't confuse the A/B sides in the pot experiment.
  • L.295: Should the degree of freedoms be (number of treatment) - 1?
  • Section 3.5 > I find the use of PCA for this particular purpose a bit unconventional. The numbers reported in Table 6 appear to be the "loadings" of each factors for the linear combinations in PCA. They do not directly indicate the influences of each variable per se. It looks like the authors combine morphological and physiological data from all treatments in one table for PCA, which probably does not tell us about the effects. Correlations in Table 5 should be sufficient, or multiple regressions would be a better test for the question of variable importance.
  • L.408: I personally am not very familiar with "comprehensive scores" from PCA. Please describe how to calculate these numbers and how to interpret them in the Method section.
Comments on the Quality of English Language

The manuscript is generally well-written. Several punctuational and editorial errors are found in the manuscript, but they should be relatively easy to fix.

Author Response

Thank you very much for giving me valuable comments, in response to the comments you gave me, I have made the following changes to the thesis: 1, in the research methodology and analysis of the results added the interpretation and calculation process of principal component analysis. If you feel that it is still unable to achieve your expected goal, I will replace the principal component analysis with multiple regression analysis.2. I have improved the pictures, tables and texts.3. I have reduced the abstract of the article.
